# Vitamin D Attenuates Loss of Endothelial Biomarker Expression in Cardio-Endothelial Cells

**DOI:** 10.3390/ijms21062196

**Published:** 2020-03-22

**Authors:** Chi-Cheng Lai, Wang-Chuan Juang, Gwo-Ching Sun, Yu-Kai Tseng, Rong-Chang Jhong, Ching-Jiunn Tseng, Tzyy-Yue Wong, Pei-Wen Cheng

**Affiliations:** 1Cardiovascular Center, Kaohsiung Veterans General Hospital, Kaohsiung 81362, Taiwan; llccheng@gmail.com; 2Department of Cardiology, Kaohsiung Municipal United Hospital, Kaohsiung 80457, Taiwan; 3Department of Pharmacy and Graduate Institute of Pharmaceutical Technology, Tajen University, Pingtung 90741, Taiwan; 4Department of Emergency, Kaohsiung Veterans General Hospital, Kaohsiung 81362, Taiwan; wcchuang@vghks.gov.tw; 5Department of Business Management, National Sun Yat-Sen University, Kaohsiung 80424, Taiwan; 6Department of Anesthesiology, Kaohsiung Medical University Hospital, Kaohsiung 80756, Taiwan; gcsun39@yahoo.com.tw; 7Faculty of Medicine, College of Medicine, Kaohsiung Medical University, Kaohsiung 80708, Taiwan; 8Department of Medical Research, Chang Bing Show Chwan Health Memorial Hospital, Changhua 50544, Taiwan; iamtsengyukai@gmail.com; 9Department of Orthopedics, Show Chwan Memorial Hospital, Changhua 50008, Taiwan; 10Department of Medical Education and Research, Kaohsiung Veterans General Hospital, Kaohsiung 81362, Taiwan; jrchang@vghks.gov.tw (R.-C.J.); cjtseng@vghks.gov.tw (C.-J.T.); 11Department of Biomedical Science, National Sun Yat-Sen University, Kaohsiung 80424, Taiwan; 12International Center for Wound Repair and Regeneration, National Cheng Kung University, Tainan 70101, Taiwan

**Keywords:** cardiovascular disease, vitamin D, stretching, TGF-β1, fibrosis

## Abstract

Vitamin D is associated with cardiovascular health through activating the vitamin D receptor that targets genes related to cardiovascular disease (CVD). The human cardiac microvascular endothelial cells (HCMECs) were used to develop mechanically and TGF-β1-induced fibrosis models, and the rat was used as the isoproterenol (ISO)-induced fibrosis model. The rats were injected with ISO for the first five days, followed by vitamin D injection for the consecutive three weeks before being sacrificed on the fourth week. Results showed that mechanical stretching reduced endothelial cell marker CD31 and VE-cadherin protein expressions, as well as increased α-smooth muscle actin (α-SMA) and fibronectin (FN). The transforming growth factor-β1 (TGF-β1) reduced CD31, and increased α-SMA and FN protein expression levels. Vitamin D presence led to higher protein expression of CD31, and lower protein expressions of α-SMA and FN compared to the control in the TGF-β1-induced fibrosis model. Additionally, protein expression of VE-cadherin was increased and fibroblast-specific protein-1 (FSP1) was decreased after vitamin D treatment in the ISO-induced fibrosis rat. In conclusion, vitamin D slightly inhibited fibrosis development in cell and animal models. Based on this study, the beneficial effect of vitamin D may be insignificant; however, further investigation of vitamin D’s effect in the long-term is required in the future.

## 1. Introduction

Acute cardiomyopathy patients are commonly found to have low vitamin D [1], which has been associated with increased mortality in critically ill patients [2]. Two key enzymes, 25-hydroxylase in the liver and 1α-hydroxylase in the kidney, are essential to the activation of vitamin D. The administration of active vitamin D has been reported to confer cardiovascular protection and increased survival in clinical [3] and experimental researches [4,5]. Paricalcitol (25-dihydroxyvitamin D2), a vitamin D receptor (VDR) activator, is reported to improve cardiovascular health or survival in patients with advanced kidney disease who tend to have low vitamin D [6,7]. Deficiency in 25-hydroxyvitamin D (1,25(OH)_2_D, or vitamin D) has been associated with cardiovascular disease (CVD) risk factors, such as age, high blood pressure, obesity and diabetes [8]. The association of vitamin D with CVD risk factors have been attributed to hypertension, congestive heart failure, myocardial infarction and stroke [9]. However, the mechanism for CVD involving vitamin D is not clear.

The risk factors for CVD vary differently, therefore vitamin D’s effect is observed from different aspects. For example, atherosclerosis accumulates plaque of fats in the blood vessels, blocking the blood flow. This blockade hinders normal blood flow, increasing blood pressure as the blood pushes through the thickening walls, leading to hypertension development as the blockade continues. As atherosclerosis progresses, the effect of pressure built on the thickened walls of the blood vessels leads to extracellular matrix (ECM) secretion, such as fibronectin, for remodeling [10,11]. ECM accumulation gradually leads to fibrosis in the vascular cells. Vitamin D is involved in the renin–angiotensin II system, which regulates blood pressure through vasoconstriction when blood pressure is increased [12]. Vitamin D deficiency is also linked to fibrosis [13,14,15,16] and immune T-cell activation [17,18]. In addition, vitamin D prevents liver fibrosis through inhibition of fibrosis markers TGF-β1, collagen I and α-SMA [19]. Therefore, our previous studies demonstrated that paricalcitol ameliorated isoproterenol (ISO)-induced cardiac dysfunction and fibrosis via regulation of endothelial-to-mesenchymal cell transition (EndoMT) [20]. Vitamin D plays a role in modulating fibrosis; however, the role of vitamin D in fibrosis of cardiovascular cells is not clear.

Vitamin D binds to its receptor with high affinity, the vitamin D receptor (VDR), which belongs to the nuclear receptor family [21] and is expressed in cardiovascular cells, including vascular smooth muscle cells, endothelial cells, cardiomyocytes and immune cells [22]. The VDR is activated by VDR activators, such as calcitriol, paricalcitol and alfacalcidol, to prevent cardiac fibrosis through inhibition of left ventricular hypertrophy [23]. In this study, we proposed that vitamin D is responsible for suppressing fibrosis in cardiovascular cells. 

## 2. Results 

### 2.1. Effect of Vitamin D on the Mechanically Induced Fibrosis Model

Increased blood pressure modulates vascular cell remodeling, which consequently affects cellular function and responses. To correctly determine the impact of mechanical stress on human cardiac microvascular endothelial cells (HCMECs), the HCMECs were mechanically stretched at 15% elongation with a 1 Hz frequency for 6 h. The HCMECs orientated and aligned perpendicularly to the direction of the uniaxial force under the given mechanical strain (Figure 1A,B). The immunofluorescence assay showed that 6h stretching increased α-SMA and α-tubulin (Figure 1C–F). Mechanical stretching significantly decreased CD31 and increased α-SMA protein levels (Figure 1G–J). Vitamin D was administered to the dynamic culture to study its effect on fibrogenesis. Mechanical stimulation reduced endothelial marker VE-Cad, and vitamin D did not restore its protein expression (Figure 2A,B). However, the release of TGF-β1 was decreased in the presence of vitamin D compared to the control (Figure 2C). In addition, vitamin D slightly attenuated the decrease of VE-cad compared to the control; the increase of fibronectin (FN) in the presence of vitamin D might be due to an onset of remodeling as pressure is built up by mechanical stretching (Figure 2D). Furthermore, the vitamin D receptor VD3R was increased in vitamin D presence; the Akt pathway that is associated with VD3R activation was not affected when vitamin D was present (Figure 2E). 

### 2.2. Effect of Vitamin D on the TGF-β1-Induced Fibrosis Model

To further determine the effect of vitamin D on fibrosis, the TGF-β1-induced fibrosis model was established. The HCMECs were induced with 5 ng/mL TGF-β1 for 24 h. Results showed that CD31 was significantly decreased (Figure 3A,B), but Snail significantly increased at 5 ng/mL TGF-β1, 24 h TGF-β1 treatment (Figure 3C,D). The HCMECs were further induced with 5 ng/mL TGF-β1 for 2 and 5 days because the expression of the fibrosis marker was not significantly increased when compared to the control in the 24 h TGF-β1 treatment. α-SMA and FN were significantly increased in a time-dependent manner at 5 ng/mL after 5 days of TGF-β1 treatment (Figure 3A,B). In addition, Snail was significantly increased at 5 ng/mL TGF-β1 on Day 5 (Figure 3C,D). The immunofluorescence assay showed that vitamin D slightly increased CD31 expression compared to the TGF-β1-treated group (Figure 4A,B). In the presence of vitamin D, the CD 31 decrease was not attenuated, FN and α-SMA were not suppressed (Figure 4C,D). Furthermore, the vitamin D receptors VD3R, VEGFR and TGFβR1, receptors which have been reported to interact with vitamin D, were increased in the presence of TGF-β1 and vitamin D (Figure 4E,F). 

### 2.3. Vitamin D Slightly Attenuated Fibrosis Biomarker in an ISO-Induced Fibrosis Model

Further investigation of vitamin D’s effect on CVD led to the analysis of myocardial heart tissue. Results showed that the myocardial infarction heart expressed the fibrosis marker FSP1, which was reduced when vitamin D was administered (Figure 5A,B). The effects of vitamin D are summarized in Table 1, thereby confirming the fundamental effect of vitamin D on fibrogenesis in cardiovascular cells.

## 3. Discussions

Fibrosis is a state when tissue become fibrous as myofibroblasts accumulate excess ECM components, such as fibronectin, α-smooth muscle actin (α-SMA) and collagen I. Mechanistically, fibroblasts secrete the transforming growth factor-β1 (TGF-β1) to promote myofibroblast differentiation [24,25]. Apart from activation of fibroblast and epithelial-to-mesenchymal transition (EMT), endothelial-to-mesenchymal transition (EndoMT) is also a source of myofibroblasts [26]. The secreted TGF-β1 remains inactive in the ECM until being activated by mechanical cues in the ECM, such as the contraction force of a myofibroblast [27,28,29]. The mechanical cues can alter ECM arrangement [30], cell adhesion [31], morphology, intra-cellular cytoskeletal organization [32], gene expression, and so forth. The accumulating ECM components pose a strain on the cardiac blood vessels that lead to pressure overload. Moreover, the contraction and relaxation of the heart exert pressure on endothelial cells at the luminal surface, and a pressure overload can lead to cardiac fibrosis [33]. 

The role of vitamin D on fibrosis is observed to be beneficial; however, statistical data showed no significance. The reason might be that vitamin D affects the CVD-related symptoms, in particular, hypertension, which is discussed in this study. Based on our observations, the endothelial cell markers CD31 and VE-Cad were slightly increased when vitamin D is present (Figure 1, Figure 3 and Figure 5). Consistent with the data on fibrosis markers was that vitamin D slightly suppressed the increase of α-SMA and FN in mechanically and TGF-β-induced models, as well as FSP1 in a rat model (Figure 1, Figure 3 and Figure 5). 

Recently, vascular cells were shown to require the mechanically activated ion channel to maintain endothelial cell function and vascular architecture in hypertension-associated vascular remodeling [34,35]. A breakthrough finding in 2007 showed that 27–33% of total cardiac fibroblasts are of endothelial origin in the heart failure murine model [36]. Evidence has shown that EndMT contributes to cardiac fibrosis in streptozotocin-induced diabetes mice with cells expressing the α-SMA and FSP1 [37]. As observed from our data, the fibrosis markers expressed in HCMEC were α-SMA and FN, and FSP1 was expressed in the in vivo model (Figure 1G, Figure 3E and Figure 5D). The effect of vitamin D on the fibrosis marker increase may be resulted from the progressive state of the fibrosis. Overall, vitamin D is observed to suppress fibrosis. The accumulation of ECM during hypertension activates the remodeling process in vascular smooth muscle cells to allow blood vessel to withstand the increased blood pressure. At the moment of blood-vessel wall remodeling, the effect of vitamin D on the cardiovascular cells is not yet clarified. In our data, the vitamin D receptor VD3R was increased in presence of vitamin D (Figure 2C,D and Figure 4E,F). In addition, the VD3R increase was not significant, indicating that the vitamin D receptor was not activated. 

The limitation of this study is that the signaling pathway of vitamin D has not been examined due to the insignificance of vitamin D’s effect on cardiovascular cell fibrosis. However, the effect of vitamin D on our health remains an interesting topic, and further investigating how vitamin D is involved in cardiac fibrogenesis or before the onset of fibrogenesis would help us understand the mechanism of vitamin D on CVD.

## 4. Materials and Methods 

### 4.1. Cell Culture

The cells HCMECs were purchased from ScienCell Research Laboratories, Inc. (Catalogue #6000, Carlsbad, CA, USA). The HCMECs were maintained in endothelial cell medium (Catalogue #1001; ScienCell Research Laboratories, Inc., Carlsbad, CA, USA), supplemented with 10% fetal bovine serum (Catalogue #0025; ScienCell Research Laboratories, Inc., Carlsbad, CA, USA), 100 IU/mL penicillin and 100 μg/mL streptomycin. The HCMECs were incubated at 37 °C, with 5% CO_2_, passaged every four days, and passages from 1 to 7 were used for this study. 

### 4.2. Chemicals

Recombinant human TGF-β1 (GFH39; CellGS) was prepared in double-distilled water at stock concentration of 0.1 mg/mL. Paricalcitol or vitamin D (Zemplar) was purchased from the pharmacy counter as a 5 μg/mL stock solution containing propylene glycol (30% *v*/*v*) and ethanol (20% *v*/*v*), and was diluted in a medium to a final concentration of 1 nM. 

### 4.3. In Vitro Stretching Device

The stretching device (ATMS Boxer) was provided by the manufacturer TAIHOYA (Kaohsiung, Taiwan) for this study. The cells were cultured on polydimethylsiloxane (PDMS) membrane coated with collagen I for one day before being subjected to stretching. The PDMS with the attached cells were mounted to the stretch device using clamps that were fixed on both ends of the PDMS membrane. The cells were stretched to 15% elongation, with a frequency of 1 Hz for 6 h. 

### 4.4. Immunofluorescence Microscopy 

Cells were fixed with 4% paraformaldehyde for 20 min, washed with PBS, and then incubated with Triton X-100 for 10 min (0.5% *v*:*v*). Blocking was done by incubating with 5% (*w*:*v*) bovine serum albumin (BSA) for 30 min, washed with PBS, and then incubated with primary antibody overnight at 4 °C. Anti-α-SMA (Abcam, ab5694, 1:100, Cambridge, UK), anti-CD31 (Santa Cruz, SC1506, 1:100, Dallas, TX, USA) and vE-cadherin (Arigo-ARG11036, 1:100, Hsinchu, Taiwan) primary antibodies were used. After binding to the primary antibodies, cells were further incubated for 1h with Alexa Fluor-conjugated anti-rabbit and anti-mouse (Jackson Immunoresearch, 1:500, West Grove, PA, USA). Finally, cells were stained and mounted using the Prolong^®^Diamond Antifade Mounting Medium containing DAPI (Life Technology, Carlsbad, CA, USA). Cells were analyzed microscopically with an Olympus DP71 device by magnification of 100 times and 200 times, or the Re-scan confocal microscope (Confocal.nl, Amsterdam, The Netherlands) at 600 times magnification.

### 4.5. Western Blot

To collect cell lysates, cells were lysed in lysis buffer, and protein concentration was quantified by the Bio-Rad assay kit (Bio-Rad Laboratories, Hercules, CA, USA). The protein was separated by sodium dodecyl sulfate-polyacrylamide gel electrophoresis. After gel electrophoresis, separated proteins were transferred to the PVDF membrane and immunoblotted with primary antibodies: Collagen I GeneTex, GTX26308, CD31 Santa Cruz, SC1506, fibronectin Abcam, ab2413, Snail Cellsignal, #3879, Slug Abcam, ab27568, vE-cadherin Arigo-ARG11036, VDR3 Cellsignal, #12550s, TGF-β1R Abcam, ab31013, VEGFR Abcam, ab32152, p-Akt^S473^ Cellsignal, #3879, Akt Cellsignal, #9276s and β-actin. The signals were visualized by enhanced chemiluminescence using ECL (ThermoFisher Scientific, Waltham, MA, USA). 

### 4.6. Immunohistochemistry Assay

The paraffin-embedded sections of heart were cut at 5 μm, followed by dewaxing, and then quenching in 3% hydrogen peroxide methanol solution. Antigen retrieval in citric buffer (10 mM, pH 6.0) was blocked with 5% goat serum. The sections were further incubated in a solution of primary antibody FSP-1 (1:100; Arigo-ARG55205, Hsinchu, Taiwan) for 4 °C overnight. The next day, the sections were incubated in biotinylated secondary antibody (1:200; Vector Laboratories, Burlingame, CA, USA) for 1 h. For visualization, the tissue was incubated in DAB (Vector Laboratories, Burlingame, CA, USA) solution, counterstained with hematoxylin.

### 4.7. ELISA

The secreted TGF-β1 was measured by the ELISA kit from Cloud-Clone Corporation (Katy, TX, USA) according to the manufacturer’s instruction. Briefly, cell culture supernatant was collected from cells subjected to stretching for 6 h. The readings were measured by a microplate reader at 450 nm. 

### 4.8. Animal

To analyze the effect of vitamin D on fibrosis, a rat model was established by isoproterenol (ISO) induction. The animal preparation is approved by the Kaohsiung Veterans General Hospital institutional review board, approved serial number was VGHKS103-029, from January 2015 until December 2015, following standard guidelines for animal care, and is based on a previous method [20]. In brief, 8-week-old male Wistar–Kyoto (WKY) rats, which weighed between 250 and 350 g, were obtained from the National Science Council Animal Facility (Taipei, Taiwan), housed in the animal center at Kaohsiung Veterans General Hospital, and were provided normal rat chow (Purina, St. Louis, MO, USA) and tap water ad libitum. The control rats were given normal saline with 0.1% ascorbic acid. The ISO rats were given 2 mg/kg per day once every day for 5 days, and ISO with 200 ng vitamin D or paricalcitol thrice in a week for 3 weeks. The control, ISO and ISO + vitamin D rats were monitored for tail systolic blood pressure and echocardiography. The rats were sacrificed after 4 weeks; their hearts were extracted and sectioned for immunohistochemistry analysis.

### 4.9. Statistical Analysis

All measurements were produced at least three times under independent conditions. The results are shown as the mean ± standard error of the mean (SEM). A two-sided Student’s t test was used to compare the mean values obtained from two independent conditions: * *p* < 0.05 indicates a significant result; ** *p* < 0.01 indicates a very significant result; and *** *p* < 0.05 indicates a highly significant result. 

## 5. Conclusions 

In conclusion, vitamin D suppressed the decrease of CD31 protein expression in the TGF-β-induced fibrosis models, and slightly suppressed the decrease of VE-Cad protein expression in the mechanically and ISO-induced fibrosis models. Our data showed that vitamin D is beneficial for suppressing fibrogenesis. Therefore, vitamin D can be a long-term supplement for individuals having high CVD risk.

## Figures and Tables

**Figure 1 ijms-21-02196-f001:**
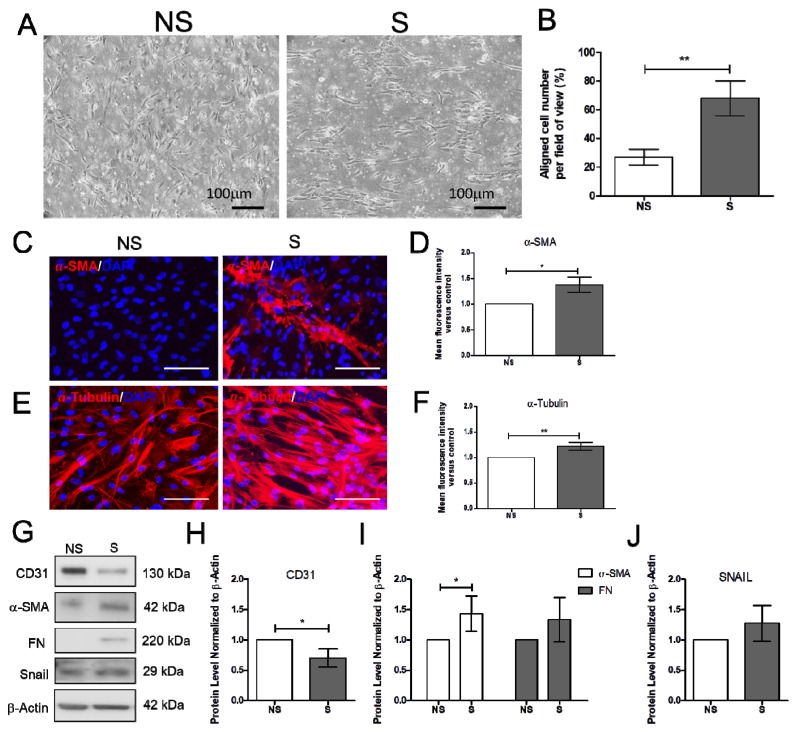
Mechanical stretching reduced CD31, but increased α-smooth muscle actin and fibronectin. (**A**,**B**) Cells orientated perpendicular to the direction of uniaxial stretching after 6 h of stretching. (**C**–**F**) Expression of α-smooth muscle actin (α-SMA), and α-tubulin analyzed by immunofluorescence assay. Scale bar = 100 μm. (**G**–**J**) Expression of endothelial cell marker CD31, fibrosis markers α-SMA and FN and the endothelial-to-mesenchymal transition marker Snail. Data are represented as at least three independent experiments. ** *p* < 0.01; * *p* < 0.05.

**Figure 2 ijms-21-02196-f002:**
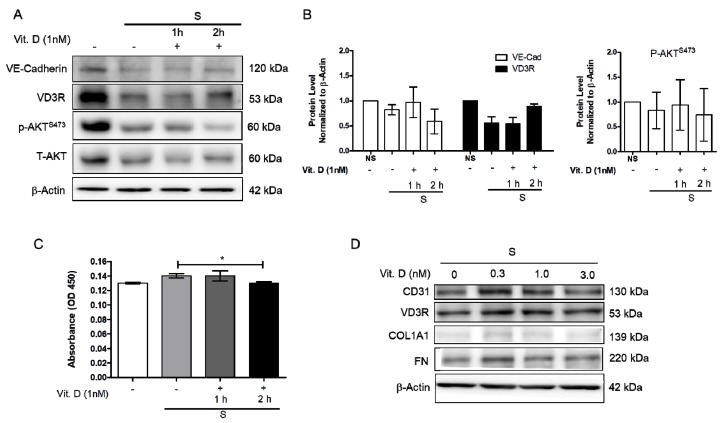
Vitamin D attenuated decrease of endothelial cell marker vascular E-cadherin protein expression. (**A**,**B**) Protein expression of endothelial marker VE-cadherin, VD3R, p-Akt^S473^ and T-Akt. (**C**) The release of TGF-β1 measured by ELISA in presence of vitamin D at 1 nM. (**D**) Protein expression of CD31, vitamin D3 receptor VD3R and fibrosis marker COL1A1, as well as fibronectin FN when 1nM vitamin D was added during the stretching. Data are represented as at least three independent experiments. * *p* < 0.05.

**Figure 3 ijms-21-02196-f003:**
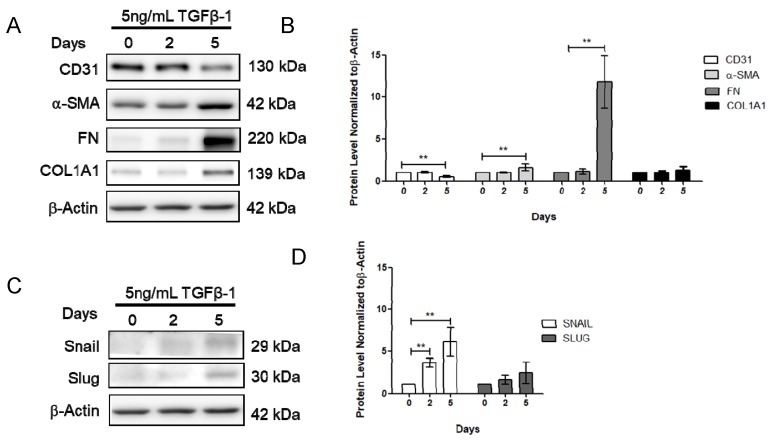
Transforming growth factor β1 reduced CD31 but increases α-smooth muscle actin and fibronectin. (**A**,**B**) Expression of CD31, α-SMA, FN and COL1A1 after 0, 2, and 5 days of TGF-β1 treatment. (**C**,**D**) Expression of Snail and Slug after 0, 2, and 5 days of TGF-β1 treatment. Data are represented as at least three independent experiments. ** *p* < 0.01; * *p* < 0.05.

**Figure 4 ijms-21-02196-f004:**
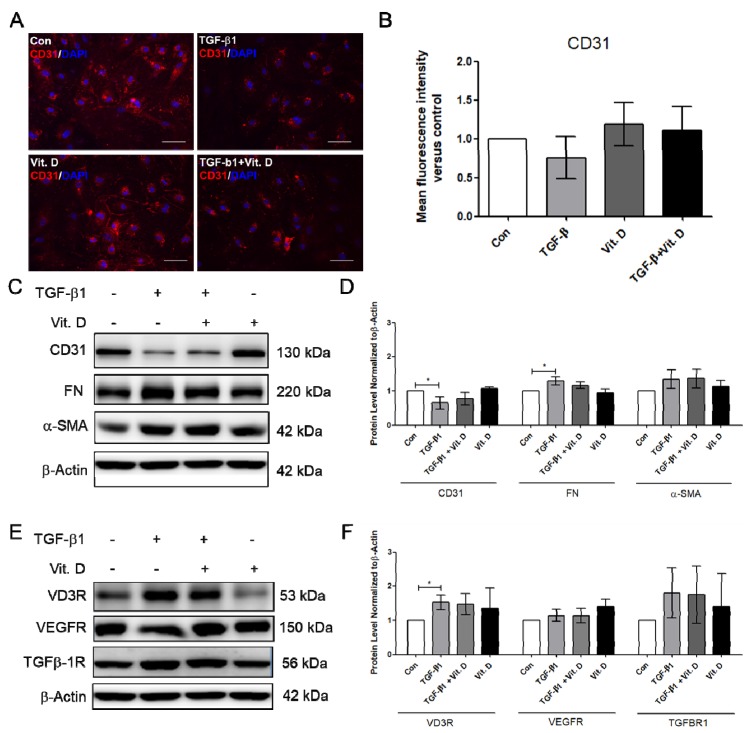
Vitamin D slightly attenuated decrease of endothelial cell marker CD31. (**A**,**B**) Immunofluorescence assay for CD31 in the TGF-β1-induced fibrosis. Scale bar = 100 μm. (**C**,**D**) Protein expression of CD31, FN, and α-SMA in presence of vitamin D at 1nM. (**E**,**F**) Protein expression of VD3R, growth factor receptors VEGFR and TGF-β1R. Data are represented as at least three independent experiments. * *p* < 0.05.

**Figure 5 ijms-21-02196-f005:**
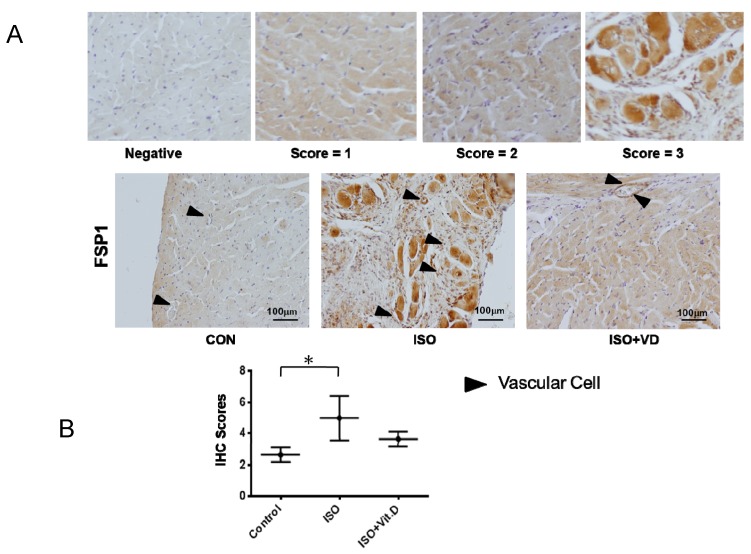
Vitamin D attenuated fibronectin specific protein-1 protein expression in fibrosis animal model. (**A**,**B**) Immunohistological analysis for fibronectin-specific protein-1 (FSP1) in the heart section of ISO-induced rats. For scoring the FSP1 expression, the score was divided into 0, 1, 2 and 3 from negative to positive expressions. The black arrow showed the blood vessels. Scale bar = 100 μm. Data are represented as at least three independent experiments. * *p* < 0.05.

**Table 1 ijms-21-02196-t001:** Summary of vitamin D’s effect in the mechanically, chemically and isoproterenol-induced cardiovascular fibrosis models.

Approaches	+S	+ TGFβ-1	+ISO
Dynamics	Cyclic stretching	Static	Rat, in vivo
Duration for induction	6 Hours	5 Days	4 Weeks
Fibrosis markers	FN, α-SMA	FN, α-SMA	FSP-1
Vitamin D effect	VE-Cad decrease attenuated	CD31 decrease attenuated	FSP-1 decreased

+S: Induction of fibrosis markers by mechanical stimulation, stretching. +TGF-β1: Induction of fibrosis markers by adding TGF-β1. +ISO: Induction of fibrosis markers by isoproterenol injection into rat, a myocardial infarction model.

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
