# Peer review of "Vitamin D Attenuates Loss of Endothelial Biomarker Expression in Cardio-Endothelial Cells"

_ijms, 2020, doi:10.3390/ijms21062196_

Round 1
Reviewer 1 Report
Lai and colleagues postulate that vitamin D attenuates the loss of endothelial marker expression in cardiovascular cells. They used human cardiac microvascular endothelial cells, stretched them or treated with TGF beta and rats with isoproterenol-induced fibrosis.
Mechanical stretching reduced endothelial cell CD31 and VE-cadherin expression, increased -smooth muscle actin and fibronectin. Vitamin D slightly attenuated VE-cadherin decrease. TGF beta treatment reduced CD31, and increased -smooth muscle actin and fibronectin. In this model, vitamin D slightly attenuated the decrease of CD31 and slightly decreased -SMA and FN 37 expression. In the ISO-induced rat model, vitamin D attenuated the decrease of VE-cadherin expression. They concluded that vitamin D can be a long-term supplement for individuals having high CVD risk.
The title is misleading. They used a endothelial cell line instead of a mix of caridovascular cells.
Multiple errors in spelling and grammar make it very hard to read the manuscript.
The introduction is very short and superficial. Statements like « Vitamin plays a role in modulating fibrosis… » are not acceptable.
Methods : 1% penicillin and streptomycin – concentration is missing.
“Two-fold of the HCMECs orientated and aligned perpendicularly to the direction of uniaxial force 144 under the given mechanical strain (Fig. 1A-1B)” is unclear.
“The CD31 was decreased, whereas both -SMA and -tubulin were increased after 6h stretching (Fig. 1C-1F). The expression of CD31 was decreased, -SMA was significantly increased after stretching (Fig. 1G-1I) – is the same message. The authors want to refer to immunocytochemistry and Western Blot data?
Figure 1 C-F – the images do not correspond to the quantification.
Figure 2, except for TGF release, nothing seems to be significant in response to Vitamin D treatment?
Figures 3 and 4: Five days of TGF beta stimulation is very long and thus, it is questionable whether these effects are specific. Furthermore, Vitamin D seems not to have any significant effect, which is in contrast to the Abstract.
Line 197: “Further investigation of vitamin effect…”. Besides Vitamin D there are plenty of other vitamin effects. Therefore, investigation of vitamin effect on cardiovascular disease is by far out of the scope of the presented study.
Figure 5A is completely non-specific staining, but not VE-cadherin expression of endothelial cells. It remains a mystery why it is less in the Iso sample.
Table 1 is unclear. According to the quantifications, there are nearly no significant Vitamin D effects. Thus, the line in the Table is wishful thinking.
Discussion: “Fibrosis is a disease state in which tissue become fibrous…” meaning?
Discussion and conclusion are only very limited related to the data in the present manuscript.
Author Response
Point-to-point Comments (by Reviewer 1) and Responses
Lai and colleagues postulate that vitamin D attenuates the loss of endothelial marker expression in cardiovascular cells. They used human cardiac microvascular endothelial cells, stretched them or treated with TGF beta and rats with isoproterenol-induced fibrosis.
Mechanical stretching reduced endothelial cell CD31 and VE-cadherin expression, increased a-smooth muscle actin and fibronectin. Vitamin D slightly attenuated VE-cadherin decrease. TGF beta treatment reduced CD31, and increased a-smooth muscle actin and fibronectin. In this model, vitamin D slightly attenuated the decrease of CD31 and slightly decreased a-SMA and FN 37 expression. In the ISO-induced rat model, vitamin D attenuated the decrease of VE-cadherin expression. They concluded that vitamin D can be a long-term supplement for individuals having high CVD risk.
Point #1
The title is misleading. They used a endothelial cell line instead of a mix of caridovascular cells. Vitamin D Attenuates Loss of Endothelial Biomarker Expression in Cardioendothelial Cells.
Response to point#1
Thanks reviewer for the comment. We have already modified the title from 〝Vitamin D Attenuates Loss of Endothelial Biomarker Expression in Cardiovascular Cells〞to〝Vitamin D Attenuates Loss of Endothelial Biomarker Expression in cardio-endothelial Cells〞in the revised manuscript. (P1, Line 2-3).
Point #2
Multiple errors in spelling and grammar make it very hard to read the manuscript.
Response to point#2
Thanks for the valuable comments of the reviewer. We highly appreciate the valuable suggestions from reviewers about grammar and typographical error in the manuscript. We have taken reviewers’ suggestions to amend the description’s error and this manuscript was also edited for proper English language by native English speaking editors at Nature Publishing Group, and the certificate key is 6605-332D-C645-068F-68E8.
Point #3
The introduction is very short and superficial. Statements like « Vitamin plays a role in modulating fibrosis… » are not acceptable.
Response to point#3
Thanks for the critical suggestions of the reviewer. We have reorganized the introduction in the revised manuscript as in the following:
Acute cardiomyopathy are commonly found to have low vitamin D level [1], which has been associated with increased mortality in critically ill patients [2]. Two key enzymes, 25-hydroxylase in the liver and 1α-hydroxylase in the kidney, are essential to the activation of vitamin D. The administration of active vitamin D has been reported to confer cardiovascular protection and increased survival in clinical [3] and experimental researches [4, 5]. Paricalcitol (25-dihydroxyvitamin D2), a vitamin D receptor (VDR) activator, is reported to improve cardiovascular health or survival in patients with advanced kidney disease who tend to have low vitamin D [6, 7]. Deficiency in 25-hydroxyvitamin D (1,25(OH)2D, or vitamin D) has been associated with cardiovascular disease (CVD) risk factors such as age, high blood pressure, obesity and diabetes [8]. The association of vitamin D with CVD risk factors have been attributed to hypertension, congestive heart failure, myocardial infarction and stroke [9]. However, the mechanism for CVD involving vitamin D is not clear.
The risk factors for CVD vary differently, therefore vitamin D effect is observed from different aspects. For example, atherosclerosis accumulates plaque of fats in the blood vessel, blocking the blood flow. The blockade hinders normal blood flow, increasing blood pressure as the blood pushes through the thickening walls, leading to hypertension development as the blockade continues. As atherosclerosis progresses, the effect of pressure built on the thickened walls of blood vessels leads to extracellular matrix (ECM) secretion such as fibronectin for remodeling [10, 11]. ECM accumulation gradually leads to fibrosis in the vascular cells. Vitamin D is involved in the renin-angiotensin II system which regulates blood pressure through vasoconstriction when blood pressure increase [12]. Vitamin D deficiency is also linked to fibrosis [13, 14],[15, 16] and immune T cell activation[17, 18]. In addition, vitamin D prevents liver fibrosis through inhibition of fibrosis markers TGF-b1, collagen I, and a-SMA [19]. Therefore, our previous studies demonstrated that paricalcitol ameliorated isoproterenol (ISO)-induced cardiac dysfunction and fibrosis via regulation of endothelial-to-mesenchymal cell transition (EndoMT) [20]. Vitamin plays a role in modulating fibrosis; however, the role of vitamin D in fibrosis of cardiovascular cells is not clear.
Vitamin D binds to its receptor with high affinity, vitamin D receptor (VDR) which belongs to nuclear receptor family [21] and is expressed in cardiovascular cells including vascular smooth muscle cells, endothelial cells, cardiomyocytes, and immune cells [22]. The VDR is activated by VDR activators such as calcitriol, paricalcitol and alfacalcidol to prevent cardiac fibrosis through inhibition of left ventricular hypertrophy [23]. In this study, we proposed that vitamin D is responsible for suppressing fibrosis in cardiovascular cells. (P2, Line45-76)
Point #4
Methods : 1% penicillin and streptomycin – concentration is missing.
“Two-fold of the HCMECs orientated and aligned perpendicularly to the direction of uniaxial force 144 under the given mechanical strain (Fig. 1A-1B)” is unclear.
Response to point#4
Thanks for the suggestions of the reviewer. We have clarified it (100 IU/ml penicillin and 100 g/ml streptomycin) in the methods of revise manuscript. (P2, Line 82).
Point #5
“The CD31 was decreased, whereas both a-SMA and a-tubulin were increased after 6h stretching (Fig. 1C-1F). The expression of CD31 was decreased, a-SMA was significantly increased after stretching (Fig. 1G-1I) – is the same message. The authors want to refer to immunocytochemistry and Western Blot data?
Response to point#5
Indeed, the error has been notified and this statement has been corrected in the revised manuscript:
Immunofluorescence assay showed that stretching increased a-SMA and a-tubulin after 6h stretching (Fig. 1C-1F). Mechanical stretching significantly decreased CD31 and increased a-SMA protein levels (Fig. 1G-1J).
Point #6
Figure 1 C-F – the images do not correspond to the quantification.
Response to point#6
Fig. 1C-1D represents a-SMA protein expression comparison between control (No stretch) and 15% Stretch, 1.52-Fold increase (P<0.05; N=3) in 15% Stretch when compared to No stretch.
Fig. 1E-11F represents tubulin protein expression comparison between control (No stretch) and 15% Stretch, 1.22-Fold increase (P<0.01; N=3) in 15% Stretch when compared to No stretch.
Point #7
Figure 2, except for TGF release, nothing seems to be significant in response to Vitamin D treatment?
Response to point#7
Indeed, we appreciate the comment by the reviewer regarding Figure 2. As superficial as it has been for the beneficial effect of vitamin D on human`s health, our data show that vitamin D does not affect the protein expression of endothelial cells` biomarker in the mechanically-stressed endothelial cells. However, the effect of vitamin D might be instantaneous; therefore we suspected its effect on the active TGF-b1 production. The increase in vitamin D receptor (VD3R) is found to be consistent with the TGF-b1 production increase; nevertheless, the protein level of VD3R showed no significance (Fig. 2B-2C).
Point #8
Figures 3 and 4: Five days of TGF beta stimulation is very long and thus, it is questionable whether these effects are specific. Furthermore, Vitamin D seems not to have any significant effect, which is in contrast to the Abstract.
Response to point #8
Indeed, TGF-b1 treatment requires longer time to induce the anticipated fibrosis biomarkers: a-SMA and FN. Nevertheless, the TGF-b1 effect on fibrogenesis in numerous cell types have been well-known.
Regarding the contrast found in the previous manuscript, the following corrections have been made to eliminate the contrasting statements (Line 31-41) in the “Abstract” section:
“Results showed that mechanical stretching reduced endothelial cell marker CD31 and VE-cadherin protein expressions, increased a-smooth muscle actin (a-SMA) and fibronectin (FN). The transforming growth factor b1 (TGF-b1) reduced CD31, increased a-SMA and FN protein expression levels. Vitamin D presence led to higher protein expression of CD31, lower protein expressions of a-SMA and FN compared to control in the TGF-b1-induced fibrosis model. Additionally, protein expression of VE-cadherin was increased and fibroblast-specific protein (FSP1) was decreased after vitamin D treatment in the ISO-induced fibrosis rat. In conclusion, vitamin D slightly inhibited fibrosis development in cell and animal models. Based on this study, the beneficial effect of vitamin D may be insignificant; however, further investigation of vitamin D effect for long-term is required in the future.”
Point #9
Line 197: “Further investigation of vitamin effect…”. Besides Vitamin D there are plenty of other vitamin effects. Therefore, investigation of vitamin effect on cardiovascular disease is by far out of the scope of the presented study.
Response to point#9
Thanks for the suggestions of the reviewer. We appreciate about our written have a misreading and thus could not follow it. We have already rewritten the sentences to〝Further investigation of vitamin D effect on CVD led to the analysis of myocardial heart tissue.〞(P8, Line206)
Point #10
Figure 5A is completely non-specific staining, but not VE-cadherin expression of endothelial cells. It remains a mystery why it is less in the Iso sample.
Response to point#10
We really appreciate the comment by the reviewer. The VE-cadherin staining shown in Fig. 5A might be due to non-specific staining, even after having optimized our experimental procedures. Therefore, we referred to the immunohistochemistry (IHC) for further answer. The IHC results show that fibronectin specific protein-1 (FSP-1) is slightly decreased when the animal is treated with vitamin D (Fig. 5C-5D).
Point #11
Table 1 is unclear. According to the quantifications, there are nearly no significant Vitamin D effects. Thus, the line in the Table is wishful thinking.
Response to point#11
This comment is highly appreciated. Based on our study, the effect of vitamin D is considered “beneficial” towards attenuation of fibrogenesis, however, might not be “significant”.
Point #12
Discussion: “Fibrosis is a disease state in which tissue become fibrous…” meaning?
Discussion and conclusion are only very limited related to the data in the present manuscript.
Response to point#12
Thanks for your critical comments. We have already modify the sentence from 〝Fibrosis is a disease state in which tissue become fibrous…〞to the following in the “Discussion” section on Page 9:
“Fibrosis is a state when tissue become fibrous as myofibroblasts accumulate excess ECM components such as fibronectin, a-smooth muscle actin (a-SMA), and collagen I. Mechanistically, fibroblasts secrete the transforming growth factor-b1 (TGF-b1) to promote myofibroblasts differentiation [24, 25]. Apart from activation of fibroblast and epithelial-to-mesenchymal transition (EMT), endothelial-to-mesenchymal transition (EndoMT) is also the source of myofibroblasts [26]. The secreted TGF-b1 remains inactive in the ECM until being activated by mechanical cue in the ECM such as contraction force of myofibroblast [27-29]. The mechanical cues can alter ECM arrangement [30], cell adhesion [31], morphology, intra-cellular cytoskeletal organization [32], gene expression, and so forth. The accumulating ECM components pose strain on the cardiac blood vessels that lead to pressure overload. Moreover, the contraction and relaxation of heart exert pressure on endothelial cells at the luminal surface, and pressure overload can lead to cardiac fibrosis [33].
The role of vitamin D on fibrosis is observed to be beneficial; however, statistical data showed no significance. The reason might be that vitamin D affects the CVD-related symptoms, in particular, hypertension which is discussed in this study. Based on our observation that the endothelial cell markers CD31 and VE-Cad were slightly increased when vitamin D is present (Fig. 1, 3, 5). Consistent with the data on fibrosis markers was that vitamin D slightly suppressed the increase of a-SMA and FN in mechanically- and TGF-b-induced models, as well as FSP1 in rat model (Fig. 1, 3, 5).
Recently, vascular cells were shown to require the mechanically-activated ion channel to maintain endothelial cell function and vascular architecture in hypertension-associated vascular remodeling [34, 35]. A breakthrough finding in 2007 showed that 27-33% of total cardiac fibroblasts are of endothelial origin in the heart failure murine model [36]. Evidence has shown that EndMT contributes to cardiac fibrosis in streptozotocin-induced diabetes mice with cells expressing the a-SMA and FSP1 [37]. As observed from our data, the fibrosis markers expressed in HCMEC were a-SMA and FN, and FSP1 was expressed in the in vivo model (Fig. 1G, 3E, 5D). The effect of vitamin D on the fibrosis marker increase may be resulted from progressive state of fibrosis. Overall, vitamin D is observed to suppress fibrosis. The accumulation of ECM during hypertension activates remodeling process in vascular smooth muscle cells to allow blood vessel to withstand the increased blood pressure. At this moment of blood vessel walls` remodeling, the effect of vitamin D on the cardiovascular cells is not yet clarified. In our data, the vitamin D receptor VD3R was increased in presence of vitamin D (Fig. 2C, 2D, 4E, 4F). In addition, the VD3R increase was not significant, indicating that vitamin D receptor was not activated. The limitation of this study is that the signaling pathway of vitamin D has not been examined due to the insignificance of vitamin D effect on the cardiovascular cell fibrosis. However, the effect of vitamin D on our health remains an interesting topic, investigating how vitamin D is involved in cardiac fibrogenesis or before the onset of fibrogenesis helps us understand the mechanism of vitamin D on CVD.” (P9, Line 230-233; P10, Line 234-267)
References:
- Ng, L. L.; Sandhu, J. K.; Squire, I. B.; Davies, J. E.; Jones, D. J., Vitamin D and prognosis in acute myocardial infarction. Int J Cardiol 2013, 168, (3), 2341-6.
- Quraishi, S. A.; Bittner, E. A.; Blum, L.; McCarthy, C. M.; Bhan, I.; Camargo, C. A., Jr., Prospective study of vitamin D status at initiation of care in critically ill surgical patients and risk of 90-day mortality. Crit Care Med 2014, 42, (6), 1365-71.
- Tamez, H.; Zoccali, C.; Packham, D.; Wenger, J.; Bhan, I.; Appelbaum, E.; Pritchett, Y.; Chang, Y.; Agarwal, R.; Wanner, C.; Lloyd-Jones, D.; Cannata, J.; Thompson, B. T.; Andress, D.; Zhang, W.; Singh, B.; Zehnder, D.; Pachika, A.; Manning, W. J.; Shah, A.; Solomon, S. D.; Thadhani, R., Vitamin D reduces left atrial volume in patients with left ventricular hypertrophy and chronic kidney disease. Am Heart J 2012, 164, (6), 902-9 e2.
- Meems, L. M.; Cannon, M. V.; Mahmud, H.; Voors, A. A.; van Gilst, W. H.; Sillje, H. H.; Ruifrok, W. P.; de Boer, R. A., The vitamin D receptor activator paricalcitol prevents fibrosis and diastolic dysfunction in a murine model of pressure overload. J Steroid Biochem Mol Biol 2012, 132, (3-5), 282-9.
- Bae, S.; Yalamarti, B.; Ke, Q.; Choudhury, S.; Yu, H.; Karumanchi, S. A.; Kroeger, P.; Thadhani, R.; Kang, P. M., Preventing progression of cardiac hypertrophy and development of heart failure by paricalcitol therapy in rats. Cardiovasc Res 2011, 91, (4), 632-9.
- Teng, M.; Wolf, M.; Lowrie, E.; Ofsthun, N.; Lazarus, J. M.; Thadhani, R., Survival of patients undergoing hemodialysis with paricalcitol or calcitriol therapy. N Engl J Med 2003, 349, (5), 446-56.
- Thadhani, R.; Appelbaum, E.; Chang, Y.; Pritchett, Y.; Bhan, I.; Agarwal, R.; Zoccali, C.; Wanner, C.; Lloyd-Jones, D.; Cannata, J.; Thompson, T.; Audhya, P.; Andress, D.; Zhang, W.; Ye, J.; Packham, D.; Singh, B.; Zehnder, D.; Manning, W. J.; Pachika, A.; Solomon, S. D., Vitamin D receptor activation and left ventricular hypertrophy in advanced kidney disease. Am J Nephrol 2011, 33, (2), 139-49.
- Gouni-Berthold, I.; Krone, W.; Berthold, H. K., Vitamin D and cardiovascular disease. Current vascular pharmacology 2009, 7, (3), 414-22.
- Danik, J. S.; Manson, J. E., Vitamin d and cardiovascular disease. Current treatment options in cardiovascular medicine 2012, 14, (4), 414-424.
- Harvey, A.; Montezano, A. C.; Lopes, R. A.; Rios, F.; Touyz, R. M., Vascular Fibrosis in Aging and Hypertension: Molecular Mechanisms and Clinical Implications. The Canadian journal of cardiology 2016, 32, (5), 659-668.
- Lan, T. H.; Huang, X. Q.; Tan, H. M., Vascular fibrosis in atherosclerosis. Cardiovascular pathology : the official journal of the Society for Cardiovascular Pathology 2013, 22, (5), 401-7.
- Selvin, E.; Najjar, S. S.; Cornish, T. C.; Halushka, M. K., A comprehensive histopathological evaluation of vascular medial fibrosis: insights into the pathophysiology of arterial stiffening. Atherosclerosis 2010, 208, (1), 69-74.
- Shi, Y.; Liu, T.; Yao, L.; Xing, Y.; Zhao, X.; Fu, J.; Xue, X., Chronic vitamin D deficiency induces lung fibrosis through activation of the renin-angiotensin system. Scientific Reports 2017, 7, (1), 3312.
- Repo, J. M.; Rantala, I. S.; Honkanen, T. T.; Mustonen, J. T.; Koobi, P.; Tahvanainen, A. M.; Niemela, O. J.; Tikkanen, I.; Rysa, J. M.; Ruskoaho, H. J.; Porsti, I. H., Paricalcitol aggravates perivascular fibrosis in rats with renal insufficiency and low calcitriol. Kidney international 2007, 72, (8), 977-84.
- Tan, X.; Li, Y.; Liu, Y., Paricalcitol attenuates renal interstitial fibrosis in obstructive nephropathy. Journal of the American Society of Nephrology : JASN 2006, 17, (12), 3382-93.
- Park, J. W.; Bae, E. H.; Kim, I. J.; Ma, S. K.; Choi, C.; Lee, J.; Kim, S. W., Renoprotective effects of paricalcitol on gentamicin-induced kidney injury in rats. American Journal of Physiology-Renal Physiology 2009, 298, (2), F301-F313.
- Pincikova, T.; Paquin-Proulx, D.; Sandberg, J. K.; Flodström-Tullberg, M.; Hjelte, L., Vitamin D treatment modulates immune activation in cystic fibrosis. Clinical & Experimental Immunology 2017, 189, (3), 359-371.
- González-Mateo, G. T.; Fernández-Míllara, V.; Bellón, T.; Liappas, G.; Ruiz-Ortega, M.; López-Cabrera, M.; Selgas, R.; Aroeira, L. S., Paricalcitol Reduces Peritoneal Fibrosis in Mice through the Activation of Regulatory T Cells and Reduction in IL-17 Production. PLOS ONE 2014, 9, (10), e108477.
- Abramovitch, S.; Sharvit, E.; Weisman, Y.; Bentov, A.; Brazowski, E.; Cohen, G.; Volovelsky, O.; Reif, S., Vitamin D inhibits development of liver fibrosis in an animal model but cannot ameliorate established cirrhosis. American Journal of Physiology-Gastrointestinal and Liver Physiology 2015, 308, (2), G112-G120.
- Lai, C. C.; Liu, C. P.; Cheng, P. W.; Lu, P. J.; Hsiao, M.; Lu, W. H.; Sun, G. C.; Liou, J. C.; Tseng, C. J., Paricalcitol Attenuates Cardiac Fibrosis and Expression of Endothelial Cell Transition Markers in Isoproterenol-Induced Cardiomyopathic Rats. Crit Care Med 2016, 44, (9), e866-74.
- Norman, P. E.; Powell, J. T., Vitamin D and Cardiovascular Disease. Circulation Research 2014, 114, (2), 379.
- Ni, W.; Watts, S. W.; Ng, M.; Chen, S. C.; Glenn, D. J.; Gardner, D. G., Elimination of vitamin D receptor in vascular endothelial cells alters vascular function. Hypertension 2014, 64, (6), 1290-1298.
- Panizo, S.; Barrio-Vazquez, S.; Naves-Diaz, M.; Carrillo-Lopez, N.; Rodriguez, I.; Fernandez-Vazquez, A.; Valdivielso, J. M.; Thadhani, R.; Cannata-Andia, J. B., Vitamin D receptor activation, left ventricular hypertrophy and myocardial fibrosis. Nephrology, dialysis, transplantation : official publication of the European Dialysis and Transplant Association - European Renal Association 2013, 28, (11), 2735-44.
- Aisagbonhi, O.; Rai, M.; Ryzhov, S.; Atria, N.; Feoktistov, I.; Hatzopoulos, A. K., Experimental myocardial infarction triggers canonical Wnt signaling and endothelial-to-mesenchymal transition. Dis Model Mech 2011, 4, (4), 469-83.
- Okayama, K.; Azuma, J.; Dosaka, N.; Iekushi, K.; Sanada, F.; Kusunoki, H.; Iwabayashi, M.; Rakugi, H.; Taniyama, Y.; Morishita, R., Hepatocyte growth factor reduces cardiac fibrosis by inhibiting endothelial-mesenchymal transition. Hypertension 2012, 59, (5), 958-65.
- Kovacic, J. C.; Mercader, N.; Torres, M.; Boehm, M.; Fuster, V., Epithelial-to-mesenchymal and endothelial-to-mesenchymal transition: from cardiovascular development to disease. Circulation 2012, 125, (14), 1795-808.
Reviewer 2 Report
The manuscript titled "vitamin D attenuates loss of endothelial biomarker expression in cardiovascular cells" by Lai and colleagues reports on a very interest and important topic. The study appears to be well design and appropriate in length. However, there are a couple of major and minor concerns regarding the manuscript that diminish its potential impact. Those concerns are provided below. Major 1. While this topic is of scientific and medical importance, I am left wondering how these results translate to human conditions in vivo. Specifically, how the administered dose of vitamin D in this study translates to supplementation in humans. 2. This is a very interesting study, but unfortunately, at times it was difficult to follow and comprehend what was being delivered to the reader. With that said, Table 1 was very helpful in bringing the manuscript together. Minor 1. Please spell out abbreviations at first use. 2. General grammar edits throughout the manuscript would improve the readability and ease the comprehension of the results. Also, the manuscript would benefit from strong proofreading. There are too many edits to list here.Author Response
Point-to-point Comments (by Reviewer 2) and Responses
Review The manuscript titled "vitamin D attenuates loss of endothelial biomarker expression in cardiovascular cells" by Lai and colleagues reports on a very interest and important topic. The study appears to be well design and appropriate in length. However, there are a couple of major and minor concerns regarding the manuscript that diminish its potential impact. Those concerns are provided below.
Point #1
Major 1. While this topic is of scientific and medical importance, I am left wondering how these results translate to human conditions in vivo. Specifically, how the administered dose of vitamin D in this study translates to supplementation in humans.
Response to Point#1
Vitamin D deficiency is regarded as a worldwide health problem that probably affects not only musculoskeletal health but also a wide range of acute and chronic diseases [1]. Therefore, the assessment of vitamin D status and the treatment of vitamin D deficiency are considered to be important issues of interest to public health [2]. Although 1,25-dihydroxyvitamin D (1,25(OH)2D) is the active, hormonal form of vitamin D, its precursor 25-hydroxyvitamin D (25OHD) is the generally accepted indicator of vitamin D status. So far, circulating 1,25(OH)2D levels have received relatively little attention, except in chronic kidney disease (CKD) patients. This is, at least in part, due to the fact that the half-life of 1,25(OH)2D in the circulation is only a few hours and the fact that circulating 1,25(OH)2D levels are considered to be tightly regulated within a narrow range [2, 3]. Due to its picomolar concentrations and its lipophilic nature, 1,25(OH)2D represents the most difficult challenge of all the steroid hormones to the analytical biochemist with respect to quantification [3].
This study aims to analyze the effect of the active form of vitamin D which is short-lived, and the results show that it has beneficial effect on reducing the fibrosis biomarker (fibronectin specific protein-1; FSP-1) in vivo. Nevertheless, the beneficial effect could not be augmented further. This result reflects that fact the vitamin D has beneficial effect in vivo in our body, although the effect may not be highly significant.
Point #2
- This is a very interesting study, but unfortunately, at times it was difficult to follow and comprehend what was being delivered to the reader. With that said, Table 1 was very helpful in bringing the manuscript together.
Response to Point#2
Thanks for the valuable comments of the reviewer. We highly appreciate the valuable suggestions from reviewers about grammar and typographical error in the manuscript. We have taken reviewers’ suggestions to amend the description’s error and this manuscript was also edited for proper English language by native English speaking editors at Nature Publishing Group, and the certificate key is 6605-332D-C645-068F-68E8.
Point #3
Minor 1. Please spell out abbreviations at first use.
Response to Point#3
Thanks for the comment. We have noticed the abbreviations and ensure that all abbreviations are spelt out at it`s first use in the revised manuscript.
Point #4
- General grammar edits throughout the manuscript would improve the readability and ease the comprehension of the results. Also, the manuscript would benefit from strong proofreading. There are too many edits to list here.
Response to Point#4
Thanks for the valuable comments of the reviewer. We have taken reviewers’ suggestions to amend the description’s error and this manuscript was also edited for proper English language by native English speaking editors at Nature Publishing Group, and the certificate key is 6605-332D-C645-068F-68E8.
References:
- Zittermann, A.; Gummert, J. F., Nonclassical vitamin D action. Nutrients 2010, 2, (4), 408-25.
- Holick, M. F., Vitamin D deficiency. N Engl J Med 2007, 357, (3), 266-81.
- Schmidlin, S.; Bioteau, C.; Detavernier, M.; Couturier, P.; Gavazzi, G., [Prevalence of vitamin D deficiency in elderly patients hospitalized in geriatric units]. Presse Med 2010, 39, (2), 271-2.
Round 2
Reviewer 1 Report
The manuscript is significantly improved. I'm still not convinced by the quality of the VE-Cadherin staining in Figure 5.
Author Response
Response to the reviewer`s comment:
We really appreciate the comment by the reviewer. The VE-cadherin staining shown in the previous Fig. 5A might be due to non-specific staining, even after having optimized our experimental procedures. Therefore, we referred to the immunohistochemistry (IHC) for further answer. The IHC results show that fibronectin specific protein-1 (FSP-1) is slightly decreased when the animal is treated with vitamin D (Fig. 5A-5B). As a result, the Figure 5A-5B has been regarded as in conclusive and data is not shown in the revised manuscript.
Reviewer 2 Report
The authors did a nice job responding to the review comments. It is a little discouraging, however, that the authors did not discuss 1,25(OH)D levels in relation to 25(OH)D levels, but rather went into a long explanation of vitamin D metabolism. and different metabolite.
Author Response
Response to reviewer`s comment:
Thanks for reviewer`s suggestion. We are apologize about have misled the reviewer`s comment about〝that the authors did not discuss 1,25(OH)D levels in relation to 25(OH)D levels.〞
The biologically active form of vitamin D is 1,25-dihydroxyvitamin D (1,25(OH)2 D). Measurement of serum levels of 1,25(OH)2 D should be considered upon suspicion of deficiency or excess of this form of vitamin. The major circulating form of vitamin D is 25-hydroxyvitamin D (25(OH)D); therefore, the total serum 25(OH)D level is currently considered the best indicator of vitamin D supply to the body from cutaneous synthesis and nutritional intake. Measurement of serum 25-hydroxyvitamin D (25[OH]D) is the best test to determine vitamin D status. Vitamin D insufficiency, levels of 25(OH)D, are interpreted as 21-29 ng/mL (52.5-72.5 nmol/L) or < 20 ng/mL (< 50 nmol/L) as vitamin D deficiency [1]. In consequence, recommended treatment for vitamin D–deficient adults is as 50,000 IU of vitamin D2 or D3 once weekly for 8 weeks or 6000 IU/day of vitamin D2 or D3 for 8 weeks. When the serum 25(OH)D level exceeds 30 ng/mL, a total amount of 1500-2000 IU/day might be applied for treatment [2]. Deficiency in 25-hydroxyvitamin D (1,25(OH)2D, or vitamin D) has been associated with cardiovascular disease (CVD) risk factors such as age, high blood pressure, obesity and diabetes [3]. The association of vitamin D with CVD risk factors have been attributed to hypertension, congestive heart failure, myocardial infarction and stroke [4].
Vitamin D insufficiency is highest among people who are elderly, institutionalized, or hospitalized. In the United States, 60% of nursing home residents [5] and 57% of hospitalized patients [6] were found to be vitamin D deficient. However, vitamin D insufficiency is not restricted to the elderly and hospitalized population; several studies have found a high prevalence of vitamin D deficiency among healthy, young adults. A study determined that nearly two thirds of healthy, young adults in Boston were vitamin D insufficient at the end of winter [7].
Vitamin D status may fluctuate throughout the year, with the highest serum 25(OH)D level occurring after the summer and the lowest serum 25(OH)D concentrations after winter. A study by Shoben at el demonstrated that mean serum 25(OH)D concentrations can vary as much as 9.5 ng/mL. Factors such as male sex, higher latitude, and greater physical activity levels were found to be associated with greater differences in serum 25(OH)D concentrations in winter and summer [8].
References
- Hollis, B.W.; Wagner, C.L. Normal serum vitamin d levels. N Engl J Med 2005, 352, 515-516; author reply 515-516.
- Lukaszuk, J.M.; Luebbers, P.E. 25(oh)d status: Effect of d3 supplement. Obes Sci Pract 2017, 3, 99-105.
- Gouni-Berthold, I.; Krone, W.; Berthold, H.K. Vitamin d and cardiovascular disease. Current vascular pharmacology 2009, 7, 414-422.
- Danik, J.S.; Manson, J.E. Vitamin d and cardiovascular disease. Current treatment options in cardiovascular medicine 2012, 14, 414-424.
- Elliott, M.E.; Binkley, N.C.; Carnes, M.; Zimmerman, D.R.; Petersen, K.; Knapp, K.; Behlke, J.M.; Ahmann, N.; Kieser, M.A. Fracture risks for women in long-term care: High prevalence of calcaneal osteoporosis and hypovitaminosis d. Pharmacotherapy 2003, 23, 702-710.
- Thomas, M.K.; Lloyd-Jones, D.M.; Thadhani, R.I.; Shaw, A.C.; Deraska, D.J.; Kitch, B.T.; Vamvakas, E.C.; Dick, I.M.; Prince, R.L.; Finkelstein, J.S. Hypovitaminosis d in medical inpatients. N Engl J Med 1998, 338, 777-783.
- Tangpricha, V.; Pearce, E.N.; Chen, T.C.; Holick, M.F. Vitamin d insufficiency among free-living healthy young adults. Am J Med 2002, 112, 659-662.
- Shoben, A.B.; Kestenbaum, B.; Levin, G.; Hoofnagle, A.N.; Psaty, B.M.; Siscovick, D.S.; de Boer, I.H. Seasonal variation in 25-hydroxyvitamin d concentrations in the cardiovascular health study. Am J Epidemiol 2011, 174, 1363-1372.
This manuscript is a resubmission of an earlier submission. The following is a list of the peer review reports and author responses from that submission.
Round 1
Reviewer 1 Report
Lai et al tested the effect of vitamin D on several models of fibrosis, namely a mechanical-induced and a TGF-beta induced fibrosis model in human cardiac microvascular endothelial cells, and a isoproterenol-induced model in rats. Overall, in the presence of vitamin D (typically at 1 nM), there is a slight but mostly insignificant suppression of selected fibrosis markers in these models. The premise of the investigation is interesting. Unfortunately, no conclusive information was found in the current study, as many changes reported are too small to be convincing. I am afraid publication in IJMS would probably demand more insights than this study can provide, therefore I would recommend rejection of this article.
Here are some of the points for the authors to note:
Conclusion in Table 1 contradicts some of the statements the authors made in text, eg in L147-148 “Mechanical stimulation reduced endothelial marker VE-Cad, and vitamin D did not restore its protein expression” and in L178-179 “In the presence of vitamin D, CD31 decrease was not attenuated…”. This is probably due to some differences did not achieve statistical significance. If the results are not clear and unambiguous even in the controlled and relatively straight-forward cell-based assays, it will be very difficult to support an argument for any clinically relevance of the observations. L46: I think 25-hydroxyvitamin D and 1,25(OH)2D are two different chemical entities, the former is calcidiol and the latter is calcitriol. L58: spelling of increase L105 to L108: Maybe put the supplier of antibodies and catalogue number in brackets Figure 1: the NS vs S label may be intuitive to some but it will still be helpful if it’s spelled out in full in the legend. Figure 2D: A bar chart depicting the quantification of 2D would be useful. As it is, this is the only Western blot without a corresponding quantification information displayed.Author Response
Comments and Suggestions for Authors
Lai et al tested the effect of vitamin D on several models of fibrosis, namely a mechanical-induced and a TGF-beta induced fibrosis model in human cardiac microvascular endothelial cells, and a isoproterenol-induced model in rats. Overall, in the presence of vitamin D (typically at 1 nM), there is a slight but mostly insignificant suppression of selected fibrosis markers in these models. The premise of the investigation is interesting. Unfortunately, no conclusive information was found in the current study, as many changes reported are too small to be convincing. I am afraid publication in IJMS would probably demand more insights than this study can provide, therefore I would recommend rejection of this article.
Here are some of the points for the authors to note:
Conclusion in Table 1 contradicts some of the statements the authors made in text, eg in L147-148 “Mechanical stimulation reduced endothelial marker VE-Cad, and vitamin D did not restore its protein expression” and in L178-179 “In the presence of vitamin D, CD31 decrease was not attenuated…”. This is probably due to some differences did not achieve statistical significance. If the results are not clear and unambiguous even in the controlled and relatively straight-forward cell-based assays, it will be very difficult to support an argument for any clinically relevance of the observations.
Point #1
L46: I think 25-hydroxyvitamin D and 1,25(OH)2D are two different chemical entities, the former is calcidiol and the latter is calcitriol.
Response to point#1
The comment is greatly appreciated. Apologize for the typing error. The term “25-hydroxyvitamin D” is corrected to “1,25-hydroxyvitamin D” in the revised manuscript: Deficiency of 1,25-hydroxyvitamin D (1,25(OH)2D, or vitamin D) has been associated with cardiovascular disease (CVD) risk factors such as age, high blood pressure, obesity and diabetes [1].
Point #2
L58: spelling of increase.
Response to point#2
Vitamin D is involved in the renin-angiotensin II system which regulates blood pressure through vasoconstriction when blood pressure increases.
Point #3
L105 to L108: Maybe put the supplier of antibodies and catalogue number in brackets.
Response to point#3
We thank reviewer for the comments. Indeed, the indication is more appropriate when full name of the company and place of origin are stated. We have included the company name, city, state and country of origin in the revised manuscript in the following subsections: 2.1 Cell culture; 2.2 Chemicals; 2.4 Immunofluorescence assay; 2.5 Western blot; 2.6 Immunohistochemistry assay; and 2.7 ELISA. The corrections made are highlighted in red using the “tracking” tool in Microsoft Word.
Point #4
Figure 1: the NS vs S label may be intuitive to some but it will still be helpful if it’s spelled out in full in the legend.
Response to point#4
Thanks reviewer for the comment. The definition for labeling of NS and S have been added in the figure legends for Figure 1 and 2 in the revised manuscript. The statement: “Abbreviations: NS: No stretch; S: Stretch” has been added to legend of Figure 1 and 2.
Point#5
Figure 2D: A bar chart depicting the quantification of 2D would be useful. As it is, this is the only Western blot without a corresponding quantification information displayed.
Response to point#5
The comment by reviewer is highly appreciated. The different concentration of vitamin D was used to evaluate its impact on CD31 protein level; however, there was no significant dose-dependent trend. The Figure 2D does not have three independent values to be calculated statistically, therefore, there is no corresponding quantification bar graph.
Submission Date
30 July 2019
Date of this review
05 Aug 2019 10:13:30

Reviewer 2 Report
The article submitted by Tzyy-Yue Wong, Pei-Wen Cheng and Co-workers, entitled “Vitamin D Attenuates Loss of Endothelial Biomarker Expression in Cardiovascular Cells” presents a set of very interesting data introducing effect of vitamin D on fibrosis progression in cardiovascular cells.
The introduction is concise and well brings into the topic of the paper and the study overall is coherent and comprehensive. To figure out the effect of vitamin D treatment/supplementation the Authors applied three experimental models of fibrosis: (i) mechanically-induced fibrosis by stretching the HCMECs, (ii) pharmacological – TGF-β1-induced fibrosis and (iii) animal – ISO-induced fibrosis. The methods for validation the hypothesis are properly chosen and described well, but the description of animal model of fibrosis in presented without sufficient detail. How many rats were used for the study? How the used vitamin D concentration reflect to its intake by the human organism? Is the concentration of vitamin D used in all three fibrosis model reachable in the in vivo conditions and how it goes with the long term treatment and toxicity of the compound? The studies performed, are properly designed and controlled, replicated suitable. The presented data are of high quality, however, Figure 5D partially covers the caption. Statistical analysis of data is performed properly. The conclusions reached, are consistent with the presented data, but the summarizing sentence suggesting “a long-term supplementation with vitamin D for individuals having high CVD risk” (presented in Abstract and in the Conclusions) should a bit modified and polished. The Authors should have in mind the toxicity of vitamin D in case of overdosing. Some minor typing errors and grammar mistakes should be removed.
I do recommend the publication of the paper as suitable for International Journal of Molecular Sciences after including the indicated comments.
Author Response
Comments and Suggestions for Authors
The article submitted by Tzyy-Yue Wong, Pei-Wen Cheng and Co-workers, entitled “Vitamin D Attenuates Loss of Endothelial Biomarker Expression in Cardiovascular Cells” presents a set of very interesting data introducing effect of vitamin D on fibrosis progression in cardiovascular cells.
The introduction is concise and well brings into the topic of the paper and the study overall is coherent and comprehensive. To figure out the effect of vitamin D treatment/supplementation the Authors applied three experimental models of fibrosis: (i) mechanically-induced fibrosis by stretching the HCMECs, (ii) pharmacological – TGF-β1-induced fibrosis and (iii) animal – ISO-induced fibrosis. The methods for validation the hypothesis are properly chosen and described well, but the description of animal model of fibrosis in presented without sufficient detail.
Point #1
How many rats were used for the study?
Response to point#1
The comments by reviewer are highly appreciated. The experiment described in Figure 5 used 6 animals for each group: Control, ISO, and ISO with vitamin D added. The experiments did not yield highly significant result between the different groups, therefore, the number of animals used are not raised. The approval ID for our animal study is: IACUC Approval Number:vghks-104-A012.
Point#2
How the used vitamin D concentration reflect to its intake by the human organism?
Point #3
Is the concentration of vitamin D used in all three fibrosis model reachable in the in vivo conditions and how it goes with the long term treatment and toxicity of the compound?
Response to point#2 & 3
The concentration of active form is not routinely measured; however, serum level of the precursor 25-hydroxyvitamin D has been reported to be 30 ng/ml. If the 25-hydroxyvitamin D serum level is below 30 ng/ml, the person is considered vitamin D deficient [1]. In our experiment, final concentration is 1 nM. For a 10 cm plate, 10 ml of completed medium was used. Another previous article indicated that serum level of 1,25-hydroxyvitamin D in patients with sepsis was 13.6 pg/ml and below [2].
From the equation Molar= (No. of moles)/(Total Volume in Litres) whereas molecular weight of 25-hydroxyvitamin D is 416.6, the final concentration of 1,25-hydroxyvitamin D (active form) used was 0.4 ng/ml, or 400 pg/ml.
Indeed, the level of vitamin D administered in the experiment is higher and may not be reasonable for in vivo application. Theoretically, the long-term usage of vitamin D as supplement will be lower than 400 pg/ml.
Point #4
The studies performed, are properly designed and controlled, replicated suitable. The presented data are of high quality, however, Figure 5D partially covers the caption. Statistical analysis of data is performed properly.
Response to point#4
Thanks for the comment. The distance between the Figure 5D and the caption (also known as figure legend) has been spaced out more with additional one-line spacing distance in the revised manuscript.
Point #5
The conclusions reached, are consistent with the presented data, but the summarizing sentence suggesting “a long-term supplementation with vitamin D for individuals having high CVD risk” (presented in Abstract and in the Conclusions) should a bit modified and polished. The Authors should have in mind the toxicity of vitamin D in case of overdosing.
Response to point#5
The comment is greatly appreciated. Indeed, the statement on the beneficial effect of vitamin D is too strong. We would like to modify the statements:
In Abstract section:1.
“Vitamin D is associated with cardiovascular health through activating the vitamin D receptor which targets genes related to cardiovascular disease (CVD), and vitamin D deficiency has been linked to fibrosis. However, the mechanism related to vitamin D in cardiac fibrosis is unclear.” The statement in Red has been added to tone and polish the original statement in the revised manuscript.
2.
“Therefore vitamin D can be a long-term supplementation with vitamin D for individuals having high CVD risk.” Has been removed.
In Conclusions section:
1.
“Our data showed that vitamin D is beneficial for suppressing fibrogenesis. Therefore, vitamin D can be a long-term supplement for individuals having high CVD risk.” Is modified to
“This study provides insight into the effect of vitamin D in fibrogenesis suppression and potential clinical application in the future.”
Point #6
Some minor typing errors and grammar mistakes should be removed.
I do recommend the publication of the paper as suitable for International Journal of Molecular Sciences after including the indicated comments.
Response to point#6
The points is well-received, we have fully revised the manuscript and is highlighted in Red using the “Tacking” tool in Microsoft Word.
Submission Date
30 July 2019
Date of this review
13 Aug 2019 14:00:56
References
[1] Ishimura E, Nishizawa Y, Inaba M, Matsumoto N, Emoto M, Kawagishi T, et al. Serum levels of 1,25-dihydroxyvitamin D, 24,25-dihydroxyvitamin D, and 25-hydroxyvitamin D in nondialyzed patients with chronic renal failure. Kidney International 1999;55:1019-27.
[2] Nguyen HB, Eshete B, Lau KHW, Sai A, Villarin M, Baylink D. Serum 1,25-dihydroxyvitamin D: an outcome prognosticator in human sepsis. PloS one 2013;8:e64348-e.

Reviewer 3 Report
there are no specific comments
Author Response
Response to the reviewer:
Thank you very much for the comments, we really appreciate them and have made necessary corrections in the revised manuscipt.

Reviewer 4 Report
in this manuscript, the authors made an attempt to understand the effect of Vit. D on Cardiac microvascular endothelial cells. The results presented doesn't support the conclusions and not even the title. I have several major concerns.
In Fig 2C, the levels of TGFb are reported. Is it active or inactive TGFb? Since the whole manuscript focussed on TGFb induced EndoMT/Fibrosis, the authors must determine the levels of canonical and non-canonical TGFb signalling following treatment with Vit.D On what basis the Vit.D concentration (1nM) was used? Sometimes the authors switched to 3nM. Interestingly, the 3nM even decreased CD31 expression in Fig 2D. This is completely contrasting to the authors conclusions and to the title! Is the concentration of Vit.D used equal to vIt.D supplementation that people take on regular basis? In Fig 1, the authors investigated the effect of Vit. D on CD31 and other markers after 6h. What happens to these markers after 24h? It is very interesting to SMA expression after 6 h in Fig 1C. Usually SMA expression takes more time in several models published so far. How come the SMA expression is increased after 6h? The authors must present the data at other time points starting from 1h in this model. Although the authors determined several markers of EndoMT, they didn't mention about it in the results except in the discussion. The authors must show the phenotype of the cells after addition of Vit.D and TGFb. In Fig 3C, the blot is not convincing at all. It must be repeated.In Fig 4A, the authors show no effect of Vit.D on CD31 expression in the immunofluroscence analyses. The authors must show higher maginifciation images. Minor: There are several typos throughout the manuscript.